# Anionic Copolymers with Different Charge Densities for Regulating the Properties of Cement Pastes

**DOI:** 10.3390/ma15217629

**Published:** 2022-10-30

**Authors:** Nanxiao Gao, Jian Chen, Min Qiao, Guangcheng Shan, Jingzhi Wu, Qianping Ran

**Affiliations:** 1State Key Laboratory of High Performance Civil Engineering Materials, Jiangsu Sobute New Materials Co., Ltd., Nanjing 211103, China; 2Bote New Materials Taizhou Co., Ltd., Taizhou 225474, China; 3Jiangsu Key Laboratory of Construction Materials, School of Material Science and Engineering, Southeast University, Nanjing 211189, China

**Keywords:** copolymer, charge density, adsorption, rheological, cement paste, viscosity-modifying admixture

## Abstract

Self-compacting concrete (SCC) is an extremely flowable concrete, which increases the probability of segregation and bleeding. Viscosity-modifying admixtures (VMAs) have been developed to improve the stability of SCC. Synthetic polymer VMAs have excellent water solubility and stability, and can be easily chemically prepared and modified. In this work, a series of copolymers based on anionic 2-acrylamido-2-methyl-propanesulfonic acid (AMPS) and nonionic N, N-dimethyl acrylamide (DMAA), with similar molecular weights but different charge densities, were prepared. The effect of the charge density of the anionic polymers on the fluidity, rheological property, and adsorption behavior of the cement pastes was investigated. The action mechanism of the polymers was discussed. The results indicate that the charge density of anionic polymer VMAs is of great significance for the development of cost-effective SCCs with good rheological properties.

## 1. Introduction

Self-compacting concrete (SCC) is an extremely flowable concrete that exhibits unique properties of self-leveling, self-deairing, and good coherence [1,2]. After filling into the formwork, SCCs do not need to be compacted laboriously, because they have flowability; they can fill out the specified formwork and compact themselves. A very dense concrete structure can be obtained, which is advantageous for the durability of concrete. Moreover, a smooth surface can be obtained, which meets the requirements of architectural concrete [3]. In order to achieve the self-compacting requirement, a higher dosage of superplasticizer is required, which increases the probability of segregation and bleeding, and then decreases the workability and adhesiveness of concretes. As a consequence, viscosity-modifying admixtures (VMAs) have been developed to improve the stability of SCC [4].

A VMA is a water-soluble polymer having a high molecular weight. The commonly used VMAs, including natural polymers (walan gum and curdlan), semi-synthetic polymers (hydroxypropyl cellulose), and synthetic polymers (polyacrylamide and polyethylene glycol), have been widely reported in previous studies [5,6,7,8,9,10,11]. Compared with traditional natural and semi-synthetic polymers, synthetic polymers have excellent water solubility and stability, and can be easily chemically prepared and modified. They have been widely used in the oil well cementing industry, but are still rarely investigated for the effects of their chemical structures on the properties of fresh cement pastes, and the action mechanism in cement pastes has also not been extensively studied.

In this work, a series of copolymers based on anionic 2-acrylamido-2-methyl-propanesulfonic acid (AMPS) and nonionic N, N-dimethyl acrylamide (DMAA), with similar molecular weights but different charge densities, was prepared by free radical solution polymerization. The effect of the charge density of the polymers on the fluidity, rheological property, and adsorption behavior of the cement pastes was investigated. The action mechanism of the polymers was discussed. The results indicate that the AMPS–DMAA copolymers exhibit the anionic polyelectrolyte property in cement pastes, and they can adsorb on the positively charged surfaces of cement particles via electrostatic attraction; the specific properties, such as fluidization, water retention, and anti-bleeding, were thus developed. The charge density of anionic polymer VMAs is of great significance for the development of cost-effective SCCs with good rheological properties.

## 2. Materials and Methods

### 2.1. Materials

#### 2.1.1. Cement

The PII 52.5 Portland cement, complying with the Chinese National Standard GB 175-1999, was purchased from Jiangnan-Xiaoyetian Cement Co. Ltd. (Nanjing, China). The cement composition was determined by X-ray diffraction and Bogue analysis. Table 1 shows the characteristics and compositions of the cement.

#### 2.1.2. Chemicals

In this study, 2-acrylamido-2-methylpropanesulfonic acid (AMPS, Sinopharm Chemical Reagent Inc., Shanghai, China), N, N-dimethyl acrylamide (DMAA, Sinopharm Chemical Reagent Inc.), sodium hydroxide (Sinopharm Chemical Reagent Inc.), 2,2′-azobis (2-methylpropionamidine) dihydrochloride (V-50, Aldrich Chemical Company Inc., Milwaukee, WI, USA), and polycarboxylic superplasticizer (PCA I, Sobute New Materials Inc., Nanjing, China) were used as received. 

#### 2.1.3. Preparation and Characterization of Copolymer AMPS–DMAA

The AMPS–DMAA random copolymers with different charge densities were prepared in our laboratory by free radical solution polymerization (Figure 1). By varying the initiator (V50) concentration, polymers with different charge densities (AMPS/DMAA molar ratios m: n of 3:1, 2:1, 1:1, 1:2, and 1:3, respectively) but similar molecular weights were successfully obtained. The AMPS–DMAA was synthesized by adding water, AMPS, and DMAA into a four-neck flask, which was stirred for 15 min. Next, sodium hydroxide was added to adjust the pH to 7~9. It is important that the pH does not drop below 7, as this would cause the spontaneous homopolymerization of the AMPS monomer. After this, 20 wt% aqueous solution of sodium APMS-DMAA was produced and polymerized at 60 °C under a nitrogen atmosphere for 2 h, using V-50 as an initiator. After 2 h, the solution was heated to 80 °C and we maintained the temperature for 4 h. During the polymerization, some water was added to decrease the viscosity of the mixture. The resulting AMPS–DMAA solution was created, with a solid content of 9 wt%. Unreacted monomers were removed through dialysis and the pure polymers were obtained.

The five AMPS–DMAA copolymers with different charge densities but similar molecular weights were synthesized by varying the molar ratio of AMPS and DMAA (Table 2). The intrinsic viscosity [η] of the APMS–DMAA copolymers was determined in a 0.5 mol/L NaCl aqueous solution with an Ubbelhode capillary viscometer at 25 °C. The following equation [10,12] was used to calculate the molecular weights, Mv, of the APMS–DMAA copolymers: [η] = 1.95 × 10^−5^ Mv^0.83^ (dL/g).

### 2.2. Tests of Spread Diameter of Cement Paste

The fluidity of cement paste is tested by measuring the values of the spread diameter. The spread diameter values of the cement paste samples were measured according to the Chinese National Standard GB/T 8077-2012 13. First, 300 g of cement, 105 g of water (a water to cement weight ratio of 0.35), 1.8 g of the solution (20%) of PCA I (0.12% of cement weight), and 0.1 g of the solutions (9%) of different AMPS–DMAA copolymers were mixed. Using an automatic cement paste mixer, the mixtures were stirred at a low speed for 2 min, rested for 30 s, and then stirred at a high speed for another 1 min, respectively. The initial fluidity of the reference sample (with 0.12% PCA I, without copolymer) was required to reach the spread diameter of 260 ± 5 mm [11]. After stirring, the cement paste was immediately poured into a Vicat cone (height 60 mm, top diameter 36 mm, bottom diameter 60 mm) placed on a glass plate, and the cone was vertically removed. The resulting spread diameter of the paste was measured twice, the second measurement being taken at a 90° angle to the first, and the values were averaged to give the final spread value.

### 2.3. Tests of the Rheological Properties of Cement Pastes

Rheological measurements were carried out on a controlled stress rheometer (R/SP-SST, Broolfield, New York, NY, USA) using a helix-type geometry. The fresh cement paste was placed into the cell and subjected to a 6 s pre-shearing cycle, which was applied at a shear rate of 100 s^−1^ to break the particle agglomerates, followed by a period of zero shear for 1 min. In this period, the sample was gently stirred to mitigate the formation of preferential shear planes due to particle orientation. Then, a linearly increasing shear rate from 0.01 s^−1^ to 100 s^−1^ within 1 min was applied to obtain the up-curve of the flow test, followed by the down-curve of the flow test from 100 s^−1^ to 0.01 s^−1^ in 1 min. The down-curve data were used to analyze the rheological properties and calculate the yield stress and plastic viscosity according to the Bingham model [13]. 

### 2.4. TOC Tests

The adsorption rates of AMPS–DMAA copolymers on the cement particles were determined by Total Organic Carbon (TOC) measurement using a Multi N/C3100 TOC analyzer (Analytikjena AG, Jena, Germany). First, 100 g aqueous solution containing different types of AMPS–DMAA copolymers was added to 100 g cement power under mechanical stirring (100 rpm) at 25 °C. Part of the suspension was separated at 5, 15, 30, 45, 60, and 90 min by centrifuging at 8000 rpm for 2 min, and then the supernatant was treated with 1 M HCl solution to remove the inorganic carbon. The amount of blank organic carbon in the cement suspension itself was obtained from the same water–cement ratio of the suspension without AMPS–DMAA copolymer at different times. The amount of organic carbon in the AMPS–DMAA copolymer without cement was considered the total TOC. The organic carbon content of the AMPS–DMAA copolymer was the difference between TOC values in the presence and absence of the AMPS–DMAA copolymer. The decrease in concentration after contact with the cement was taken as the adsorption amount.

### 2.5. Zeta Potential Tests

The test procedure for zeta potential was consistent with that elsewhere reported [13] and was used to qualitatively monitor the surface charge of cement particles. The suspensions were prepared by dispersing 100 g of cement in 100 mL deionized water containing AMPS–DMAM copolymers with different charge densities under magnetic stirring (250 rpm). The zeta potential of the cement paste was determined at 20 ± 2 °C using the DT310 zeta potential analyzer (Dispersion Technology, Bedford Hills, NY, USA).

## 3. Results and Discussion 

### 3.1. Fluidity of the Cement Pastes

Figure 1 shows the fluidity of the cement pastes mixed with different AMPS–DMAA copolymers at different incubation times. The fluidity of pastes with the addition of AMPS–DMAA copolymers was lower than that of the reference sample (without copolymers), and the values of pastes for A, B, C, D, and E were approximately 218, 210, 205, 210, and 250 mm at 5 min, respectively. The fluidity increased with the increasing time, and leveled off after 30 min. The paste fluidity presented an upward trend with the molar ratio of AMPS and DMAA, whether greater than 1:1 or less than 1:1, and the values from low to high were C < D < B < A < E at the same time. All the copolymers proved to be effective in controlling the segregation of slurry, and copolymer C, with a 1:1 molar ratio of AMPS and DMAA, showed the greatest effectiveness.

The bleeding water amounts of the cement pastes were also tested (Figure 2). The cement pastes were mixed with different polymers and incubated for 90 min. It could be seen that all samples with the addition of the polymers had smaller bleeding amounts compared with the blank sample, and the polymer with the AMPS–DMAA molar ratio of 1:1 (C) gave the smallest bleeding amount. It could be also observed that the order of bleeding amount was as follows: C < D < B < A < E < blank. This is quite consistent with the order of fluidity. The results clearly suggest that all the anionic polymers can obviously decrease the bleeding of the cement pastes, and the polymer with a medium charge density can decrease the bleeding the most effectively.

### 3.2. Rheological Properties of the Cement Pastes

Pastes mixed with the AMPS–DMAA copolymers of different charge densities were prepared, and the variations in the apparent viscosities with shear rate at 5 and 90 min were as shown in Figure 3. The apparent viscosity decreases with the increase in shear rate. For the same dosage of PCA I, pastes with all the AMPS–DMAA copolymers exhibited high apparent viscosities at low shear rates and significantly lower values at greater shear rates. The paste containing copolymer C exhibited higher apparent viscosity than the other copolymers when the shear rate was below 20 s^−1^, but with the increase in shear rate, the gap between the apparent viscosities of the pastes with copolymers A, B, C, D, and E was gradually narrowed. The apparent viscosity of the paste with copolymer C at 5 min was decreased from 0.70 Pa·S at 5 s^−1^ to 0.30 Pa·S at 20 s^−1^, compared with 0.53, 0.64, 0.68, and 0.39 Pa·S at 5 s^−1^ and 0.19, 0.22, 0.28, and 0.13 Pa·S at 20 s^−1^ in the pastes with copolymers A, B, D, and E, respectively. Similar behavior was also observed in the pastes at 90 min. It is obvious that charge density is the principal factor that influences the apparent viscosity of cement pastes, but there is no direct correlation between charge density and apparent viscosity.

Figure 4a shows the time–yield stress curves of cement pastes. Yield stress was found to decrease with time, which could be attributed to the increasing fluidity with time (Figure 1). The yield stress of the paste with copolymer C was found to be higher than those of the other copolymers, and the values were C > D > B > A > E at the same incubation time. Figure 4b shows the time–plastic viscosity curves of the cement pastes. Plastic viscosity was found to increase with time, which could be attributed to the hydration of cement with time, which caused the paste to stiffen over time [11]. The plastic viscosity of the paste with copolymer C was found to be higher than those of the other copolymers, and the values were C > D > B > A > E at the same incubation time. It could be observed that, when the molar ratio of AMPS and DMAA was 1:1, both the yield stress and the plastic viscosity of the paste were higher than those of the other copolymers, and the values from high to low were C > D > B > A > E. Moreover, for the paste containing any copolymer, A, B, C, D, or E, the yield stress decreased with time and the plastic viscosity increased with time. These results may be attributed to the fact that interactions existed among copolymer–copolymer, copolymer–water, and copolymer–cement particles, which resulted in the formation of a three-dimensional network structure between the liquid phase and the cement particles, further increasing the resistance of the grout to undergo deformation and guaranteeing the overall stability of the pastes [14]. As time elapsed, interactions existing among copolymer–copolymer, copolymer–liquid phase, and copolymer–cement particles gradually diminished, and the three-dimensional network structure disappeared; the stability of the paste was thus broken. The charge density of the copolymer presents a significant effect on the properties of cement pastes.

### 3.3. Adsorption Properties of the Cement Pastes

We plot the adsorption rates of AMPS–DMAA copolymers with different charge densities in Figure 5. The saturated adsorption rate increased with the increase in charge density. The saturated adsorption rates of copolymers A and B (higher charge density) were both above 80%, whereas values for C, D, and E were around 75%, 65%, and 40%, respectively. Moreover, there was a plateau after 15 min in the adsorption rates of copolymers A and B, whereas the adsorption rates of copolymers C, D, and E increased gradually with increasing adsorption time. AMPS–DMAA copolymers exhibited a particular affinity to the surfaces of cement particles, which was strongly influenced by the charge densities of the copolymers. The anionic sulfonic acid groups of the AMPS monomer can absorb on the positively charged surface of the aluminate via electrostatic attractions; meanwhile, the anionic sulfonic acid groups can also adsorb on the negatively charged surfaces of the silicate via Ca^2+^ ion bridging as a result of complexation [15,16,17]. 

The zeta potential values of the fresh cement pastes mixed with different copolymers were also tested (Table 3). It could be observed that the addition of the AMPS–DMAA copolymers with negative charge decreased the surface potential of cement particles. The values of zeta potential decreased with the increasing negative charge density of the AMPS–DMAM copolymers. The zeta potential values were consistent with the results of TOC. The copolymer with the higher charge density presented a higher adsorption rate; thus, the value of zeta potential was lower.

### 3.4. Correlation Investigations

The fluidity of pastes versus yield stress and plastic viscosity (as shown in Figure 6) showed an anticorrelation, whether at 5 or 90 min. However, the trend shows that higher fluidity corresponded to both lower yield stress and lower plastic viscosity. Therefore, a qualitative indication of the yield stress and plastic viscosity could be obtained from the fluidity.

As shown in Figure 7, the adsorption rates of AMPS–DMAA copolymers with different charge densities from high to low were A > B > C > D > E, whereas the fluidity values from high to low were E > A > B > D > C. For the same dosage of PCA I, it became clear that the effect of the AMPS–DMAA copolymers on the dispersion performance of PCA I originated from competitive adsorption [1]. The AMPS–DMAA copolymers decreased the quantity of PCA I adsorbed on cement particles. In theory, the adsorption sites of the cement particle surface are occupied more by the AMPS–DMAA copolymer that shows a higher charge density; the adsorption amount of PCA I will be lower, and thus the fluidity of the cement paste will be lower. However, there is no corresponding relationship, whether between the adsorption rate of AMPS–DMAA copolymers and the fluidity of cement paste, or between the charge density of AMPS–DMAA copolymers and the fluidity of cement paste. The adsorption rate of the AMPS–DMAA copolymer is not the only factor affecting the fluidity of the cement paste.

A correlation between the adsorption rate and rheological parameters (yield stress and plastic viscosity) was achieved, as shown in Figure 8. The trend showed that there was no corresponding relationship between the adsorption rate and rheological parameters. The adsorption rates of AMPS–DMAA copolymers from high to low were A > B > C > D > E at 5 min, whereas the yield stress and the plastic viscosity from high to low were C > D > B > A > E. Similar behavior was also observed in the pastes at 90 min.

We note from Figure 5 that the adsorption rate of AMPS–DMAA copolymers on cement particles increased with the increasing charge density of the copolymers. Adsorption properties play an important role in controlling the fluidity and rheological parameters (yield stress and plastic viscosity) of cement pastes, but we could not find any corresponding relationship between the adsorption rate and fluidity or rheological parameters (as shown in Figure 7 and Figure 8). We plot the influence of the charge density of the AMPS–DMAA copolymer on the rheological parameters of cement filtrate in Figure 9. The yield stress values of copolymers A, B, C, D, and E were very close to each other, whereas the plastic viscosity values showed similar behavior. However, the yield stress and plastic viscosity of these copolymers present significant differences in cement pastes. It should be noted that the adsorption behavior does not exist in cement filtrate, and thus the relative adsorption of AMPS–DMAA copolymers on the surfaces of cement particles is an important factor that can reduce the fluidity and increase the yield stress and plastic viscosity of cement pastes. This phenomenon indicates that the relative adsorption is not the only factor affecting the fluidity and rheological parameters of cement pastes. 

N. Mikanovic and J. Sharman et al. [18] reported that VMAs increased the yield stress, plastic viscosity, and apparent viscosity of fresh cement-based materials through a combination of the following mechanisms [19,20,21,22].

1. Water retention: The polymer molecule absorbs and immobilizes free water molecules through hydrogen bonding; in doing so, their apparent volume increases by swelling, which increases the viscosity of the mix water. In an AMPS–DMAA copolymer, the nonionic monomer (DMAA) can adsorb and fix free water molecules because the amide group has a stronger ability to bind water through hydrogen bonding [15].

2. Polymer–particle interaction: The adsorption of polymer onto particles leads to an increased particle size and an increase in drag; furthermore, higher polymer concentrations lead to particle–particle bridging, producing a relatively rigid network [21]. In an AMPS–DMAA copolymer, the anionic monomer (AMPS) can adsorb on the surfaces of cement particles and lead to polymer–particle interaction and particle–particle bridging.

3. Polymer–polymer interaction and entanglement: Adjacent polymer chains can develop attractive forces, resulting in the formation of a gel-like network, thus further blocking the motion of water and increasing the viscosity of the whole system; at higher concentrations, the polymer molecules can entangle, resulting in a further increase in the apparent viscosity. In an AMPS–DMAA copolymer, the nonionic monomer (DMAA) has no electrostatic repulsion, which causes the polymer to easily intertwine and entangle [15].

It can be seen that the adsorption rate increases with the increasing charge density of the AMPS–DMAA copolymers, whereas the polymer–polymer interaction and entanglement decrease. The entanglement of polymer molecules also affects the consistency of cement pastes, which has an influence on the fluidity of pastes. The fluidity of cement pastes decreased with the increasing polymer–polymer entanglement. The appropriate charge density of the AMPS–DMAA copolymer results in a copolymer that can be absorbed on cement particles and facilitates the entanglement effects between copolymer molecules.

## 4. Conclusions

In summary, a series of copolymers based on anionic AMPS and nonionic DMAA, with similar molecular weights but different charge densities, were prepared. The effect of the charge density of the polymers on the fluidity, rheological property, and adsorption behavior of the cement pastes was fully investigated. The polymer with a medium charge density (AMPS/DMAA 1:1) decreased the fluidity and increased the plastic viscosity and yield stress most effectively, indicating that it can affect the properties of cement paste most strongly. The action mechanism of the polymers towards cement pastes was also discussed. The results indicate that the AMPS–DMAA copolymers exhibit an anionic polyelectrolyte property in cement pastes, and they can adsorb on the positively charged surfaces of cement particles via electrostatic attraction; the specific properties, such as fluidization, water retention, and anti-bleeding, were thus developed. The charge density of anionic polymer VMAs is of great significance for the development of cost-effective SCCs with good rheological properties.

## Data Availability

The data are contained within the article. Additional data are available upon request from the corresponding author.

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
