# Peer review of "Anionic Copolymers with Different Charge Densities for Regulating the Properties of Cement Pastes"

_materials, 2022, doi:10.3390/ma15217629_

Round 1

Reviewer 1 Report

The submitted manuscript studied the effect of the charge density of copolymers on the fresh properties of cement paste. The experimental design is satisfactory and the work falls under the scope of the journal. The are some comments to be addressed before possible acceptance of the paper. The comments are given in the attached PDF file. 

Author Response

1. Reviewer comments: Section 1 in page 1 (L35) Superplasticizer decreases the workability?

Our response: We appreciate the helpful comments from the reviewer. SCC is an extremely flowable concrete. The dosage of superplasticizer is generally higher, which increases the probability of segregation and bleeding, and then decreases the workability and adhesiveness of concretes. We have changed L32-34 into the following sentence: In order to achieve the self-compacting requirement, a higher dosage of superplasticizer is required, which increases the probability of segregation and bleeding, and then decreases the workability and adhesiveness of concretes.

2. Reviewer comments: Section 2.1.3 in page 2 (L73-74): Where are other schemes? Only one method is explained here? Section 2.4 in page 3 (L114-115): There is no explanation about different types in Sec. 2.1.3. Section 2.5 in page 4 (L129): But how? Only one scheme is explained in the synthesis section. Section 3.1 in page 4 (L134-135): Although I am not a chemistry expert, but I am wondering that does molecular weight ever change? This explanation along with Table 2 should be shifted to Sec. 2.1.3 where synthesis procedure is explained.

Our response: We appreciate the helpful comments from the reviewer. We have changed L72-76 into the following sentence: The AMPS-DMAA copolymers with different charge densities were prepared in our laboratory by free radical solution polymerization (Scheme 1). By varying the initiator (V50) concentration, the polymers with different charge densities (AMPS/DMAA molar ratios m: n of 3:1, 2:1, 1:1, 1:2 and 1:3, respectively) but similar molecular weights were successfully obtained.

We have shifted the explanation along with Table 2 (Sec. 3.1 of unrevised manuscript) to Sec. 2.1.3.

3. Reviewer comments: Section 3.2 in page 4 (L150-151): In this content based on visual inspection of bleed water from the slurry?

Our response: We appreciate the helpful comments from the reviewer. The bleeding water amounts of the cement pastes were tested. We have added the results to the article (Figure 2 and L157-165 of the revised manuscript). The cement pastes were mixed with different polymers and incubated for 90 min. It can be seen that all samples with the addition of the polymers have smaller bleeding amounts compared with the blank sample, and the polymer with the AMPS/DMAA molar ratio of 1:1 gave the smallest bleeding amount. It can be also observed that the order of bleeding amount is as follows: AMPS/DMAA 1:1 < 1:2< 2:1 < 3:1 < 1:3 < blank, which is quite consistent with the order of fluidity. The results clearly suggest that all the anionic polymers can obviously decrease the bleeding of the cement pastes, and the polymer with medium charge density can decrease the bleeding most effectively.

4. Reviewer comments: Section 3.3 in page 5 (L159): Data of control sample is not include. It is better to add it for comparison if available.

Our response: We appreciate the helpful comments from the reviewer. We have added the data of control sample (blank) to the Figure (Figure 3 of revised manuscript).

Reviewer 2 Report

In this manuscript by N.Guo et al., the authors studied the effect of adding the copolymers (AMPS-DMAA) with the various charge densities in cement. They measured the rheological properties on the cement including the copolymers the adsorption rate on the cement particle. As a result, the authors unveiled that the adsorption rate increases with the charge density and that the entanglement effect decreases with it.  The authors clearly stated almost parts in the manuscript. Moreover, the authors discussed the origin of the rheological properties based on the structure and interaction among the samples.

 The reviewer has considered that this study is worth for the sense of controlling the properties of the concrete because the mechanical properties must depend on its structure. Some points in this manuscript, however, should be modified for the readers’ understanding. Then, the authors are requested to consider the following comments on the revisions of the manuscripts.

Section 2.1.3 in page 2 (L72-84)

What is the monomer sequence in the copolymer? Random copolymer? Block copolymer?

The monomer sequence can determine the interaction (attraction or repulsion).

Section 2.2 in page 2(L87-99):

The reviewer did not understand well the experimental set up on the “fluidity test”. Then, the experimental set up should be explained in more detail. Especially, what is the reference values and what is an input in this experiment?

The reference value is written as 260 mm in L90 in page 3. The result of fluidity on the reference sample, however, change with time. Does this reference value change with time or do authors prepare another reference sample at each time?

For reviewer, it is difficult to understand the meaning of the spread length. What input is introduced in this system?  For example, force is imposed as input? If so, the authors should write the value of the force.

Figure1 in page4 and Figure2 in page5:

The fluidity in Fig.1 and the apparent viscosity in Fig.2 seem to show essentially the same properties of the samples. Is it necessity for both data to exhibit in the manuscript?

L189-193 in page 6:

The authors should explain the “interactions existed in copolymer-copolymer” and the “network structure”.

The interaction might be the net interaction among the copolymers. Its interaction is repulsive or attractive one? Moreover, in general, the net interaction among copolymers depends on the temperature and the quality of the medium. In the situation in the manuscript, the copolymers with the various charge densities are used in experiment. Does the change in charge density affect on the “interaction”?

 Moreover, network structure means gel-like one by polymers? Or network is composed of the other particles with hydrogen-bond clusters?

L284-285 in page 10:

The reviewer did not understand the relationship between the water-retention and the intertwine or the entanglement of copolymers.

L292-294 in page 10

The authors should show the reason why the gel-like network structure appears. As the reviewer pointed out before, the interaction among the copolymers depends on the medium, temperature and so on. As a result, depending on the quality of the medium and the copolymer’s mass density (and also charge density), the copolymers can show the wide spatial distribution.

Author Response

1. Reviewer comments: Section 2.1.3 in page 2 (L72-84): What is the monomer sequence in the copolymer? Random copolymer? Block copolymer? The monomer sequence can determine the interaction (attraction or repulsion).

Our response: We appreciate the helpful comments from the reviewer. The AMPS/DMAA copolymers were prepared by free radical solution polymerization in this article. They are all random copolymers.

2. Reviewer comments: Section 2.2 in page 2 (L87-99): The reviewer did not understand well the experimental set up on the “fluidity test”. Then, the experimental set up should be explained in more detail. Especially, what is the reference values and what is an input in this experiment?

For reviewer, it is difficult to understand the meaning of the spread length. What input is introduced in this system?  For example, force is imposed as input? If so, the authors should write the value of the force.

Our response: We appreciate the helpful comments from the reviewer. The fluidity of cement paste is an index to evaluate the performance of admixture. Under a certain amount of water, the fluidity depends on the water demand of the cement. The spread diameter values of the cement paste samples were measured according to the Chinese National Standard GB/T 8077-2012 13. 300 g of cement, 105 g of water (the water to cement weight ratio of 0.35). 1.8 g of the solution (20%) of PCA I (0.12% of cement weight) and 0.1g of the solutions (9%) of different AMPS/DMAA copolymers were mixed. Using an automatic cement paste mixer, the mixtures were stirred in a low speed for 2 min, rested for 30 s and then stirred in a high speed for another 1 min, respectively. The initial fluidity of reference sample (with 0.12% PCA I, without copolymer) was required to reach the spread diameter of 260±5 mm. We have added these sentences to L98-105 of the revised manuscript.

3. Reviewer comments: Section 2.2 in page 2 (L87-99): The reference value is written as 260 mm in L90 in page 3. The result of fluidity on the reference sample, however, change with time. Does this reference value change with time or do authors prepare another reference sample at each time?

Our response: We appreciate the helpful comments from the reviewer. The reference sample was mixed with PCA I, which contains many slow-release groups and can adsorb on cement particles continuously and slowly. The dispersion effect of the cement paste enhances gradually with time. Therefore, the result of fluidity on the reference sample changes with time.

4. Reviewer comments: Figure 1 in page 4 and Figure 2 in page5: The fluidity in Figure 1 and the apparent viscosity in Figure 3 seem to show essentially the same properties of the samples. Is it necessity for both data to exhibit in the manuscript?

Our response: We appreciate the helpful comments from the reviewer. Figures 1 and 2 (Figure 3 of revised manuscript) indicate different properties of the samples in cement paste. Figure 1 shows the effect of charge density of AMPS-DMAA copolymers on the fluidity of cement paste, and Figure 3 shows the effect of charge density of AMPS-DMAA copolymers on the variation of apparent viscosity of cement paste. The fluidity of cement paste is affected by the apparent viscosity, plastic viscosity and yield stress collectively. Therefore, both Figures 1 and 3 are necessary in this study.

5. Reviewer comments: L189-193 in page 6: The authors should explain the “interactions existed in copolymer-copolymer” and the “network structure”.

The interaction might be the net interaction among the copolymers. Its interaction is repulsive or attractive one? Moreover, in general, the net interaction among copolymers depends on the temperature and the quality of the medium. In the situation in the manuscript, the copolymers with the various charge densities are used in experiment. Does the change in charge density affect on the “interaction”?

Moreover, network structure means gel-like one by polymers? Or network is composed of the other particles with hydrogen-bond clusters?

Moreover, network structure means gel-like one by polymers? Or network is composed of the other particles with hydrogen-bond clusters?

L284-285 in page 10: The reviewer did not understand the relationship between the water-retention and the intertwine or the entanglement of copolymers.

L292-294 in page 10: The authors should show the reason why the gel-like network structure appears. As the reviewer pointed out before, the interaction among the copolymers depends on the medium, temperature and so on. As a result, depending on the quality of the medium and the copolymer’s mass density (and also charge density), the copolymers can show the wide spatial distribution.

Our response: We appreciate the helpful comments from the reviewer. The action mechanism of VMA molecules toward cement particles is illustrated in Figure S1 and Refs. 18–22. We have added some sentences in this part of revised manuscript.

The anionic monomer (AMPS) can adsorb on the positively charged surface of the aluminate via electrostatic attractions. In addition, the sulfo groups can be seen as the anchoring groups for adsorption of the polymers on cement particles, which can adsorb on the negatively charged surface of the silicate via the Ca2+ ions bridging as a result of the complexation between the Ca2+ and SO32-. The adsorption of the anionic polymers on the surface of cement particles leads to particle–particle bridging, which increases the yield stress and stability of the cement pastes.

The nonionic monomer (DMAA) can adsorb and fix free water molecules because the amide group has a stronger ability to bind water through hydrogen bonding. In addition, the nonionic monomer has no electrostatic repulsion which let the polymer easy to intertwine and entangle. The particle–particle bridging and polymer–polymer entanglement result in the formation of large flocs in which free water molecules may be entrapped. Therefore, both the anionic and nonionic monomer effect on the properties of fresh cement paste cooperatively. Both the water retention action and polymer–polymer entanglement increase the plastic viscosity and reduce the water loss of cement pastes. The polymer-polymer interaction depends on the content of nonionic monomers in the polymer. The higher the charge density and the lower the nonionic content of the copolymer, the weaker the polymer-polymer interaction.

Round 2

Reviewer 2 Report

In this revised manuscript by N. Gao et al., the authors nicely revised the manuscript so that it is easier for the reviewer (and some readers) to understand it. However, the authors should explain or revise the just one point. The reviewer recommends the publications after the revision.

2.2 Fluidity test

 The reviewer has not understood the definitions of fluidity and spread diameter. Hence, how “fluidity” and “spread diameter” are measured? In the reviewer’s understanding, the fluidity (or spread diameter) is mechanical response. In order to measure this, a certain force as the input should be imposed. What force is input?  As in the revised manuscript, in L103-104, the mixtures are stirred, is the stirring force input? The authors should directly answer the question “what is an input?”.

 And then, because the reviewer does not understand the definition of fluidity, the reviewer does not understand the relationship between the fluidity and the apparent viscosity. Are these completely different one?

Author Response

1. Reviewer comments: The reviewer has not understood the definitions of fluidity and spread diameter. Hence, how “fluidity” and “spread diameter” are measured? In the reviewer’s understanding, the fluidity (or spread diameter) is mechanical response. In order to measure this, a certain force as the input should be imposed. What force is input?  As in the revised manuscript, in L103-104, the mixtures are stirred, is the stirring force input? The authors should directly answer the question “what is an input?”.

And then, because the reviewer does not understand the definition of fluidity, the reviewer does not understand the relationship between the fluidity and the apparent viscosity. Are these completely different one?

Our response: We appreciate the helpful comments from the reviewer. The fluidity of cement paste is tested by measuring the values of spread diameter. The larger the value of cement paste spread diameter, the better the fluidity. The input imposed to the cement paste is the force of gravity (the tools used for the test are shown in the following figure). The apparent viscosity is one of the parameters to measure the rheological properties of cement paste. In general, the larger the value of cement paste spread diameter, the smaller the apparent viscosity. The spread diameter of cement paste is also affected by plastic viscosity and yield stress collectively.
